# A Proposal for Neurography Referral in Patients with Carpal Tunnel Syndrome Based on Clinical Symptoms and Demographic Variables of 797 Patients

**DOI:** 10.3390/diagnostics14030297

**Published:** 2024-01-30

**Authors:** Fernando Vázquez-Sánchez, Ana Isabel Gómez-Menéndez, María López-Veloso, Sara Calvo-Simal, María Carmen Lloria-Gil, Josefa González-Santos, María Nieves Muñoz-Alcaraz, Antonio José Jiménez-Vilchez, Jerónimo J. González-Bernal, Beatriz García-López

**Affiliations:** 1Clinical Neurophysiology Service, University Hospital of Burgos, 09006 Burgos, Spain; fvazsan@saludcastillayleon.es (F.V.-S.); agomm@salucastillayleon.es (A.I.G.-M.); mclloriag@ubu.es (M.C.L.-G.); 2Internal Medicine Department, University Hospital of Burgos, 09006 Burgos, Spain; mlopezv@saludcastillayleon.es; 3Research Unit, University Hospital of Burgos, 09006 Burgos, Spain; scalvo@hubu.es; 4Department of Health Sciences, University of Burgos, 09001 Burgos, Spain; jejavier@ubu.es; 5Interlevel Clinical Management Unit of Physical Medicine and Rehabilitation, Reina Sofía University Hospital, 14011 Cordoba, Spain; marian.munoz.sspa@juntadeandalucia.es; 6Maimonides Biomedical Research Institute of Cordoba (IMIBIC), Reina Sofia University Hospital, University of Cordoba, 14004 Cordoba, Spain; 7Valle de los Pedroches Hospital, 14400 Pozoblanco, Spain; antonio.jimenez.vilchez.sspa@juntadeandalucia.es

**Keywords:** carpal tunnel syndrome, electroneurography, electromyography, sensitivity, sensory conduction velocity, distal motor latency, electrodiagnosis

## Abstract

The clinical manifestation of median nerve entrapment at the carpal tunnel level is known as carpal tunnel syndrome (CTS). Electroneurography (ENG) is considered the gold standard in CTS evaluation. We conducted a retrospective study and analyzed some clinical and demographic variables, relating them to the degree of neuropathy using ENG, to better understand the role of ENG in this very common disease. We studied 816 patients referred to our service for neurographic evaluation. Their symptoms were classified as compatible with CTS (cCTS) (*n* = 646) and atypical for CTS (aCTS) (*n* = 170). A blind ENG was performed on 797 patients. Patient characteristics were coded as variables and analyzed to study whether they could predict neuropathy severity (sensory and motor involvement or grade ≥ 3 in our classification). We found a correlation between typical symptomatology, age over 50 years, male gender, positivity of Phalen’s maneuver and Tinel’s sign, and a neuropathy grade ≥ 3. We also found a correlation with CTS in the contralateral hand if the other hand showed neuropathy, despite the lack of symptoms in this hand. We propose a practical algorithm for ENG referral based on clinical symptoms, demographic factors, and neurophysiological variables.

## 1. Introduction

Compressive median nerve neuropathy through the carpal tunnel is the most common entrapment neuropathy in clinical practice [1]. The symptoms caused by this alteration constitute the so-called carpal tunnel syndrome (CTS) [2]. Anatomical, genetic factors, age, underlying pathology [3,4], and occupational causes [5,6] contribute to the pathophysiology of the disease. The economic burden derived from this pathology is relevant due to the expenses relating to its diagnosis, treatment, and patient medical leave [7].

The diagnosis is clinical, and the typical symptoms are paresthesia in the territory of the median nerve [8,9,10], nocturnal worsening of symptoms [9], and improvement of symptoms with the flicking of the hands [11]. The classical Phalen’s definition of CTS requires sensory manifestations in the median nerve sensory territory and positivity of Tinel’s sign and Phalen’s maneuver [12]. Other signs and maneuvers have been described for the physical examination of CTS, such as inverted Phalen’s or Durkan maneuvers [8,13,14], but they are all limited by interobserver variability.

The criteria have evolved with knowledge about the pathology without significant variations among the items used [15,16,17].

The severity of symptoms can be assessed by functional scales, allowing the classification of symptoms into mild, moderate, and severe degrees [18]. Hypoesthesia in the territory of the median nerve, loss of strength in the hands, and atrophy of the muscles of the thenar eminence are symptoms related to severe nerve lesions.

Electroneurography (ENG) of the median nerve with or without electromyography (EMG) has been considered the gold standard in the evaluation of median nerve functionality by assessing sensory or motor fiber conduction through the carpal canal [19,20,21]. Some authors have suggested that it is necessary to determine the degree of nerve involvement by neurography when considering surgical treatment, indicating that a clinical diagnosis is insufficient and needs to be verified by an “objective” test [22,23,24]. Other authors have found a positive correlation between clinical symptoms reflected in CTS-6 and ENG [16,25]. The diversity of clinical criteria and the technical differences in neurophysiological examinations make it challenging to establish a prevalence and compare results between studies. Therefore, specific diagnostic standards seem necessary [26].

CTS treatment can be divided into two categories: conservative and surgical. There is evidence for conservative treatment, such as corticosteroids and splints, in mild and moderate cases [27]. In severe cases, surgical nerve decompression should be considered [28,29]. However, the cut-off for mild, moderate, or severe cases is not always clearly established in the guidelines, and clinical manifestations and the grade of neuropathy are sometimes used indistinctly [30]. Most guidelines do not provide clear recommendations for selecting patients for treatment [31,32]. Spanish guidelines for general practitioners use imprecise terms such as “very pathological neurography” without a previous definition [31].

This study aims to evaluate the correlation between clinical and demographic variables and the presence of sensory–motor neuropathy as assessed by ENG, review the indications of neurography, and suggest a practical algorithm for ENG referral.

## 2. Materials and Methods

### 2.1. Participants

We evaluated 816 patients with suspected CTS referred for neurography to the Neurophysiology Department of the University Hospital of Burgos throughout 2018.

The inclusion criteria were (1) patients older than 15 with hand symptoms considered to be CTS by the referral doctors, (2) referral to our service for ENG evaluation, and (3) consent to have ENG performed.

Exclusion criteria for ENG analysis were (1) patients not consenting to an ENG; (2) diagnosis of polyneuropathy with prominent sensory or sensory-motor involvement, preventing the determination of the degree of median nerve compression; and (3) previous surgical decompression of the median nerve.

The data were obtained from the patient’s medical records.

### 2.2. Procedures and Assessments

An experienced neurologist evaluated all patients referred in 2018 through an anamnesis and physical examination. The complete anamnesis included patient history; current disease attributes, including time of evolution and risky work activity (repetitive movements, continuous opening and closing movements of the hand, professional vibrator use, etc.); unilateral or bilateral symptomatology; type of symptoms; and previous treatments for symptoms. Treatment with nonsteroidal anti-inflammatory drugs (NSAIDs) or other drugs was not considered because some patients used them for other reasons. Steroid infiltration was anecdotal in our series. Patients were classified according to two subgroups: “symptomatology compatible with CTS” (cCTS) and “atypical symptomatology for CTS” (aCTS). cCTS was considered for patients who fulfilled any of the following criteria: (1) sensory symptoms in median nerve territory; (2) nocturnal hand symptoms with stiffness or pain that awaken the patient; (3) sensory symptoms that worsen with manual activity and improve by flicking; and (4) data from the physical examination, Tinel’s sign, and Phalen’s maneuver [15,17]. aCTS was considered for patients with a clinical suspicion of CTS who were referred to ENG evaluation but did not fulfill any of the previous criteria and showed other symptoms, such as isolated pain in the wrist and pain in the first metacarpal joint.

ENG was performed on both hands by experienced neurophysiologists with a Carefusion-Synergy EMG 2011, San Diego, CA, USA. Of the 816 patients clinically evaluated, only 19 patients had no neurography performed on any of their hands due to the exclusion criteria (7 in the aCTS group and 12 in the cCTS group; see Figure 1). The ENG was blinded for a preliminary evaluation by the neurologist. The electrodiagnostic study (EDX) was conducted following the recommendations of the American Academy of Neurology, the American Academy of Physical Medicine and Rehabilitation, and the American Association of Electrodiagnostic Medicine, with the clarification of recommendations included by all three academies in 2002 [22]. The EDX study involved an orthodromic sensory neurography of finger III. If this was normal, a significant latency difference was sought between the median and ulnar nerves in finger IV. Motor neurography of the median nerve was also performed. EMG of the thenar muscle innervated by the median nerve was only performed in severe median entrapments. The data obtained from neurography were used to classify the level of median nerve involvement in 7 degrees (Table 1). For this purpose, the neurophysiological criteria used were those shown in Table 2. To simplify the physiological variations observed as a function of age, we divided our population into those under and over 75 years of age because the velocity values are physiologically modified to a greater extent from this age onwards [33].

The cut-off grade of moderate neuropathy was considered for those patients with sensory involvement plus the onset of motor involvement (grade ≥ 3) (Table 2).

A review of medical records was made to collect clinical data and associate them with the results of the neurographies.

The Ethics Committee of Burgos and Soria accepted the study with reference number CEIm2279 in the context of a research study on hospital management of CTS.

### 2.3. Statistical Analysis

For the statistical study, an anonymized Excel database was created and subjected to analysis using the SPSS version 25 software, with the confidence level set at 95%.

A descriptive analysis of the study population was performed. We calculated the odds ratio (OR) for the risk of having moderate neuropathy (grade ≥ 3). The variables analyzed were age, gender, typical or atypical symptomatology, unilateral or bilateral presentation of symptoms, profession considered at risk for CTS, time of evolution, and positivity of Tinel’s sign and Phalen’s maneuver.

We excluded from the hand analysis (1) hands that had previously undergone surgical median nerve decompression and (2) traumatic lesions of the median nerve at wrist level leading to CTS symptoms.

The total number of hands analyzed was 1531. There were 63 missing data points from the initial 1594 expected hands. Of those, 13 were in the aCTS group and 50 in the cCTS group.

Finally, by integrating our data and previous definitions of mild (only nocturnal), moderate (symptoms during the day), and severe (thenar atrophy) symptoms [34], we created an algorithm that can help ENG referral of CTS patients and conservative treatment.

## 3. Results

The initial population consisted of 816 consecutive patients with a mean age of 54 years (+/−14.5) (15 to 90 years).

Clinical and demographical data are presented in Table 3.

None of the patients’ symptoms had been previously classified as mild, moderate, or severe following clinical scales by their referral doctors.

After evaluation by the neurologist, 79.5% of the patients (*n* = 646) were classified as “compatible with CTS” (cCTS), and 20.5% (*n* = 170) were classified as “atypical symptomatology for CTS” (aCTS). Of the 634 cCTS patients examined by ENG, 25.8% had no degree of neuropathy, and 74.1% (*n* = 470) had some degree of neuropathy in one of the hands. Of the latter, 65.3% (*n* = 307) had a grade ≥ 3. Of the 163 aCTS patients examined by ENG, 66.87% (*n* = 119) had no degree of neuropathy, and 33.1% (*n* = 54) had some degree. Of the latter, only 23.3% (*n* = 18) presented a grade ≥ 3 (Figure 1).

Table 4 shows the grades of neuropathy for the 1531 hands tested by neurography.

After statistical analysis, presentation with typical symptomatology (classification criteria), age over 50 years, male gender, symptoms in the dominant hand, bilaterality of symptoms, and the positivity of a Tinel’s sign or Phalen’s maneuver showed statistical significance for presenting with a neuropathy grade ≥ 3 (Table 5).

The time of evolution did not show a significant correlation with the degree of neuropathy, and there were no differences between referring specialists. Regarding bilateral neuropathy, 65% of the patients who had a degree of neuropathy ≥3 in the right hand had it in the left hand, and 79% of those who had a degree of neuropathy ≥3 in the left hand also had it in the right hand, as assessed by ENG.

## 4. Discussion

We studied the demographic and clinical data of our cohort and analyzed their relationship with the ENG results, focusing on the detection of neuropathy with both sensory and motor involvement (grade 3). The variables with statistical significance for this grade of neuropathy were typical symptomatology (cCTS), male gender, age, positivity of Tinel’s sign, positivity of Phalen’s maneuver, the presence of symptoms in the dominant hand, and bilateral symptoms (Table 5).

Spanish guidelines for general practitioners propose “very pathological neurography”, thenar atrophy, or more than 12 months of clinical symptoms for surgical treatment. This ENG recommendation means that motor involvement of the nerve should be present to undergo surgery, but the guidelines do not provide a clear cut-off. On the other hand, they recommend an ENG for any patient with clinical symptoms of CTS [31]. This generalization of ENG use has led to the indiscriminate performance of this test without previous thorough clinical evaluation, sometimes leading to unnecessary overtesting [28]. American guidelines do not solve this issue [32].

ENG following CTS indication represents up to 60% of the ENG requests in our department. Having some clues as to which patients will present sensorimotor neuropathy in their ENG can be useful.

Although more women than men consulted in our series, male gender was significantly correlated with a degree of neuropathy ≥3. It is also known that age increases the risk of presenting median nerve neuropathy, and in our series, it was found that age over 50 years was related to a more severe degree of neuropathy.

Concerning the clinical presentation of CTS, 74.1% of patients considered to have cCTS had some degree of neuropathy compared to those classified as aCTS, where only 33.1% presented some degree of neuropathy. Regarding the grade of neuropathy in those groups, in the cCTS group, patients with a degree of neuropathy ≥3 (*n* = 307) constituted 65.3%, whilst they only accounted for 10.4% in the aCTS group (*n* = 18). These 18 patients represented 2.2% of the total series. This could mean an incidental finding considering that they did not complain of typical symptoms of CTS. This should lead one to think that the presence of atypical symptoms makes it necessary to look for an alternative diagnosis, not necessarily based on ENG/EMG. A preselection based on the typicity of symptoms would avoid up to 20.5% of the scans performed under this indication, with a diagnostic loss of patients with ≥3 grade of neuropathy representing only 2.2% of the total number of patients.

The positivity of Phalen’s maneuver, which is widely known and easy to apply, showed an increased risk of grade ≥ 3 neuropathy, while Tinel’s sign positivity correlated with a lower OR than that observed for Phalen’s maneuver. This could be because Tinel’s sign positivity has been related to milder degrees of involvement with greater ephaptic conduction, and it disappears as the lesion of the sensory component of the nerve progresses [32].

According to previous literature, ENG presents a sensitivity of up to 85% and a specificity of 94 to 99%, depending on the technique used [22,35]. In our series, 25.4% of patients had typical symptomatology and normal ENG according to the protocol used. Our data are similar to other published series [15,16] and could be due to an onset of entrapment in the sensory fibers of fingers I or II. Nevertheless, because the motor neurography was normal in all those cases, our therapeutic approach would not have changed even if the sensory involvement of the first finger had been assessed.

In our series, patients with no neuropathy or a mild degree of neuropathy (<3) represented 51.2% in the cCTS group and 88.9% in the aCTS group. In the analysis of hands, 44% did not present any degree of neuropathy. Furthermore, very few patients had been treated with wrist splinting. Considering these data, we think that conservative treatment by splinting is underused. Treatment by steroid infiltration could be considered anecdotal in our series.

Not many authors have previously correlated the clinical characteristics of patients with abnormalities in neurography. Patients classified clinically as “definite”, according to the Witt criteria, are more likely to show an abnormal ENG (78%) compared to patients classified as “possible” (47%) [16]. Recently, other authors have found a correlation between CTS-6 and the grade of neuropathy on ENG [25]. Given the high request for neurophysiological tests and seeking an adequate balance between diagnostic sensitivity and a therapeutic approach, we searched for a selection strategy to be able to detect neuropathies with motor involvement, recommending wrist splinting as a definite treatment in patients with a low risk of sensory–motor neuropathy or as a symptomatic treatment while ENG is performed in the rest of them. In a series of untreated patients followed clinically for two years, 23% worsened, 29% remained unchanged, and spontaneous improvement was observed in 48% of the cases [36]. It should be considered that milder involvement tends to persist over time with fluctuating symptoms without worsening neuropathy [36], making conservative treatment an option in these cases. For patients with severe CTS symptoms, the decision to undergo surgery must be considered in the first instance, and then ENG is mandatory.

We found very variable positions regarding ENG referral, with some groups considering it unnecessary prior to surgery while others used it in all clinically suspicious cases of CTS. Instead of this approach, we think it is more reasonable to consider the clinical criteria of typical CTS and at least moderate symptoms before performing ENG. ENG would establish the degree of neuropathy with a view to surgical treatment or would provide valuable information in the case of a necessary differential diagnosis [37]. Conservative treatment should be started from the beginning of clinical symptoms for all patients. There would be a potential saving of ENG, especially in patients who present an evolving course with monophasic symptoms and no tendency to relapse. Good clinical guidance can also reduce neurophysiological examination time. Otherwise, an extensive protocol of CTS diagnosis and wide differential diagnosis is necessary for each patient. Finally, if the more symptomatic hand has neuropathy on ENG, it is likely to also exist in the contralateral hand, regardless of symptoms, so ENG should be performed bilaterally.

Based both on our results and the current literature, we propose the following algorithm for ENG referral in CTS (Figure 2).

This algorithm would allow us to guide diagnosis following clinical criteria, one of the strongest predictors for pathological neurography, according to our results. It establishes a cut-off regarding symptom severity based on previous definitions [34] and helps enhance symptomatic treatment following clinical suspicion. According to a recent Cochrane review, even small benefits with splitting seem to justify its use in patients. The benefits would manifest in long-term use [39]. The algorithm could also reduce current ENG demand in patients with a low risk of sensory–motor neuropathy, and this could also have an impact by shortening the waiting time for ENG in those patients at high risk of sensory–motor neuropathy. We also provide a specific and clear cut-off for ENG findings that makes it reproducible for any clinical neurophysiology department and could help in surgical decisions. The global clinical relevance should be assessed in future studies.

However, there are some limitations, including the fact that this is a retrospective study, although the clinical evaluation was done prospectively. A selection bias also limits the study because we studied patients referred to our service, which leaves out an important population group that may never be referred for neurography in CTS. A single specialist conducted the clinical evaluation, but several clinical neurophysiologists performed the neurography assessment.

## 5. Conclusions

The referral for ENG in CTS should be based on the clinical manifestations of CTS, giving priority to patients with a high risk of sensory–motor neuropathy. The best use of ENG is achieved when it is performed on patients with typical symptomatology. Using a list of clinical diagnostic criteria improves the cost-effectiveness of neurography, minimizing tests with normal findings.

Age over 50 years, male gender, positivity of Phalen’s maneuver, positivity of Tinel’s sign, and typical symptoms increase the possibility of finding a higher degree of neuropathy. We propose a practical algorithm for guiding CTS management.

## Figures and Tables

**Figure 1 diagnostics-14-00297-f001:**
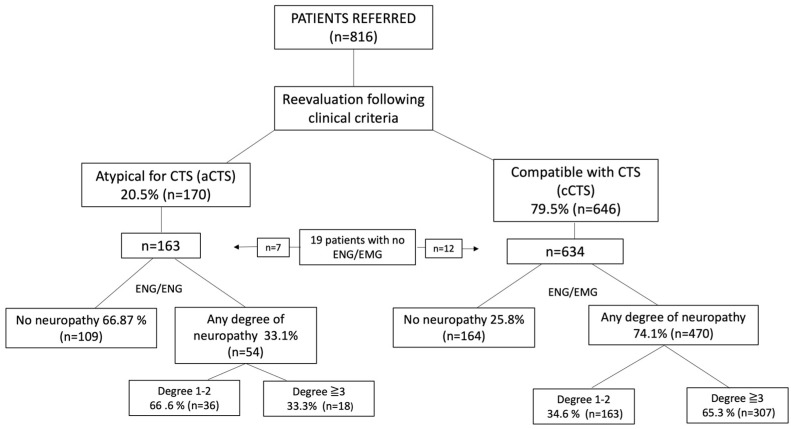
Flow chart according to clinical classification and ENG/EMG results. Patients were classified according to the higher degree of neuropathy in any of their hands.

**Figure 2 diagnostics-14-00297-f002:**
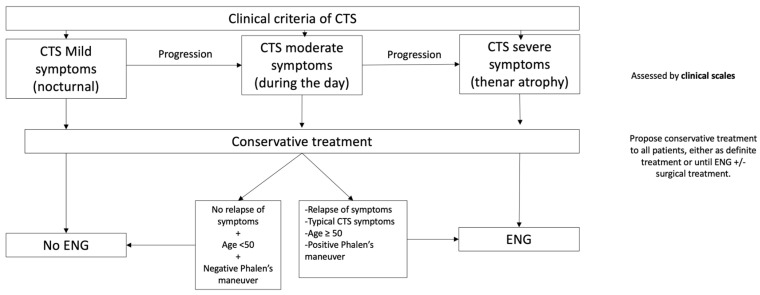
Diagnosis should be made based on clinical criteria [15,16,17]. The severity of symptoms can be assessed by any of the scales available [34,38]. Conservative treatment should be proposed to all patients, either as definite treatment or until ENG or surgery [39]. Criteria for ENG referral: OR ≥2 for neuropathy with sensory and motor involvement (≥3).

**Table 1 diagnostics-14-00297-t001:** Degrees of ENG based on the findings.

Degree	ENG/EMG FINDINGS
0: None	Normality of all parameters.
1: Incipient	Normal (SCV with objective intrapersonal abnormality (marked difference in latency in neurography after stimulation of finger IV in median nerve vs. cubital nerve or in the palm–wrist technique).
2: Mild	Decreased SCV.
3: Mild–moderate	Decreased SCV with a slight DML increase.
4: Moderate	Decreased SCV with increased DML. A light decrease in sensory potential amplitude (SA) is accepted.
5: Moderate–severe	Decreased SCV and prolonged DML with AS markedly decreased.
6: Severe	Decreased SCV with prolonged DML fulfills one of the following criteria: motor amplitude markedly decreased, absent sensory potential, or denervation in abductor pollicis brevis.
7: Very severe	Marked increase in DML with SCV markedly decreased or absent sensory potential with motor response in the surface electrode.

SCV: sensory conduction velocity; DML: distal motor latency.

**Table 2 diagnostics-14-00297-t002:** Neurophysiological criteria used in neurography for the definition of the degrees of involvement.

	<75 Years	>75 Years
Decreased SCV	<48 m/s	<44 m/s
Decreased sensory amplitude (SA)	<7 µV	<5 µV
Markedly decreased SA	<3.5 µV	<2.5 µV
Motor distal latency (MDL) slightly prolonged	>4 ms and <4.2 ms	>4.1 ms and <4.3 ms
Prolonged MDL	>4.2 ms	>4.3 ms
Very prolonged MDL	>8 ms	
Decreased motor amplitude (MA) ^1^	<4 mV	
Very decreased MA	<2 mV	

^1^ Total value or <50% compared to contralateral. m/s: meter per second; µV: microvolt; mV: millivolt; ms: millisecond.

**Table 3 diagnostics-14-00297-t003:** Demographic and clinical data of the patients.

PATIENTS	%	*n*
**Total**		816
**Gender**		
Female	70.6%	576
Male	29.4%	240
**Dexterity**		
Right-handed	96%	767
Left-handed	4%	49
**Symptom distribution**		
Bilateral	61%	498
Unilateral	39%	318
**Symptom duration**		
More than one year	55%	449
Less than one year	45%	367
**Tinel’s sign**		
Right hand	57%	465
Left hand	58%	473
**Phalen’s maneuver**		
Right hand	39%	319
Left hand	34.1%	278
**Previous conservative treatment by hand splinting**		
Symptoms compatible with CTS (cCTS)	8.2%	53
Atypical symptomatology for CTS (aCTS)	4.1%	7

**Table 4 diagnostics-14-00297-t004:** Distribution of the grade of neuropathy in the hands. There were no significant differences between degrees in the right hand compared to the left hand.

Grade of Neuropathy	Hands (*n* = 1531)	
	*n*	%
0	672	43.8
1	84	5.5
2	281	18.5
3	122	8
4	184	12
5	60	4
6	113	7.5
7	15	1

**Table 5 diagnostics-14-00297-t005:** Results of statistical univariate analysis.

VARIABLE		ENG < 3		ENG > 3		OR (Confidence Interval 95%)	*p*-Value
		n	%	n	%		
Age	<50	463	45	138	28	2.06 (1.636; 2.599)	<0.001
	50	576	55	354	72		
Gender	Female	761	73	316	64	1.525 (1.212; 1.919)	<0.001
	Male	278	27	176	36		
Symptoms in the dominant hand	No	537	52	215	44	1.393 (1.122; 1.730)	0.003
	Yes	493	48	275	56		
Bilateral symptoms	No	403	39	162	33	1.322 (1.055; 1.65)	0.015
	Yes	619	61	6329	67		
Positivity of Tinel’s sign	No	440	46	158	35	1.557 (1.236; 1.962)	<0.001
	Yes	524	54	293	65		
Positivity of Phalen’s Maneuver	No	668	70	217	49	2.412 (1.914; 3.090)	<0.001
	Yes	291	30	228	51		
Patients cCTS criteria	No	288	28	25	5	7.164 (4.685; 10.954)	<0.001
	Yes	751	72	467	95		

## Data Availability

The data presented in this study are available on request from fvazsan@saludcastillayleon.es, after authorization, previously requested to our hospital. The data are not publicly available due to the privacy policy of our institution.

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
