# Peer review of "A Proposal for Neurography Referral in Patients with Carpal Tunnel Syndrome Based on Clinical Symptoms and Demographic Variables of 797 Patients"

_diagnostics, 2024, doi:10.3390/diagnostics14030297_

Round 1

Reviewer 1 Report

Comments and Suggestions for Authors

This paper introduces a practical algorithm for the referral of electroneurography (ENG) based on a comprehensive analysis of clinical symptoms, demographic factors, and neurophysiological variables. The significance of this study is underscored by the widespread prevalence of carpal tunnel syndrome, a condition that incurs substantial healthcare costs.

The research methodology is well-structured, and the clarity of the methods employed enhances the reliability of the study. However, certain aspects require attention for improvement. Specifically, the presentation of results lacks clarity, as some findings are initially presented in the results section and then reiterated in the discussion section, creating potential confusion for readers.

Furthermore, a noteworthy observation pertains to the placement of the algorithm, a pivotal component of the study. While the algorithm represents a critical outcome, it is currently positioned at the conclusion of the discussion section. To align with conventional reporting practices, it is essential to relocate the algorithm to the results section, where it can be appropriately presented and discussed.

Addressing these concerns will enhance the overall coherence and impact of the paper, ensuring a more effective communication of the research findings and contributing to the advancement of knowledge in the field of carpal tunnel syndrome diagnosis and management.

Line 98: It is advised to provide a more explicit elucidation of the terms "cCTS" and "aCTS" for the sake of clarity. Additionally, clarification is sought on whether the aCTS group underwent any alternative diagnostic examinations or differential diagnoses.

Line 136: A typographical error has been identified in the phrase "We excluded from the analysis hand analysis." Kindly rectify this typographical error.

Line 145: The statement "had a job considered of risk for developing CTS" would benefit from further specification, and it is recommended to include a reference. The controversy surrounding whether certain occupational roles, such as computer-based work or heavy labor, contribute to CTS should be acknowledged.

Lines 196-222: The information presented in this section appears to represent results rather than discussions. To enhance clarity, it is suggested to refine the results paragraph for improved coherence and subsequently discuss the findings.

Line 266 and Algorithm: As previously emphasized, an algorithm is a consequential outcome of the study and, therefore, should be initially presented as a result. Subsequently, the algorithm can be thoroughly discussed in relation to existing literature. This adjustment will adhere to standard reporting conventions and facilitate a more comprehensive understanding of the research outcomes.

In summary, the paper demonstrates considerable significance, yet it necessitates enhanced organizational structure. Moreover, I encourage the authors to incorporate references to the American Academy of Orthopaedic Surgeons (AAOS) guidelines on carpal tunnel syndrome (https://www.aaos.org/quality/quality-programs/upper-extremity-programs/carpal-tunnel-syndrome/), a notable omission in the current reference list. An insightful discussion elucidating the authors' findings in light of the AAOS guidelines would contribute significantly to the overall scholarly discourse.

Author Response

This paper introduces a practical algorithm for the referral of electroneurography (ENG) based on a comprehensive analysis of clinical symptoms, demographic factors, and neurophysiological variables. The significance of this study is underscored by the widespread prevalence of carpal tunnel syndrome, a condition that incurs substantial healthcare costs.     

The research methodology is well-structured, and the clarity of the methods employed enhances the reliability of the study. However, certain aspects require attention for improvement. Specifically, the presentation of results lacks clarity, as some findings are initially presented in the results section and then reiterated in the discussion section, creating potential confusion for readers.

Furthermore, a noteworthy observation pertains to the placement of the algorithm, a pivotal component of the study. While the algorithm represents a critical outcome, it is currently positioned at the conclusion of the discussion section. To align with conventional reporting practices, it is essential to relocate the algorithm to the results section, where it can be appropriately presented and discussed.

Addressing these concerns will enhance the overall coherence and impact of the paper, ensuring a more effective communication of the research findings and contributing to the advancement of knowledge in the field of carpal tunnel syndrome diagnosis and management.

Line 98: It is advised to provide a more explicit elucidation of the terms "cCTS" and "aCTS" for the sake of clarity. Additionally, clarification is sought on whether the aCTS group underwent any alternative diagnostic examinations or differential diagnoses.

Thank you for you advise, we have clarified the information as follows:

Patients were classified according to two subgroups: "symptomatology compatible with CTS" (cCTS) and "atypical symptomatology for CTS" (aCTS). cCTS were considered patients fulfilling any of the following criteria: 1) Sensory symptoms in median nerve territory, 2) Nocturnal hand symptoms with stiffness or pain that awaken the patient, 3) Sensory symptoms that worsen with manual activity, and improve by flicking, 4) Data from the physical examination; Tinel's sign or Phalen's maneuver (15,17). aCTS were considered patients with a clinical suspicion of CTS referred to ENG evaluation not fulfilling any of the previous criteria, and showing other symptoms, such as isolated pain in the wrist, pain in the first metacarpal joint…

Line 136: A typographical error has been identified in the phrase "We excluded from the analysis hand analysis." Kindly rectify this typographical error.

We have corrected the text: "We excluded from the hand analysis"…

Line 145: The statement "had a job considered of risk for developing CTS" would benefit from further specification, and it is recommended to include a reference. The controversy surrounding whether certain occupational roles, such as computer-based work or heavy labor, contribute to CTS should be acknowledged.

Thank you for your suggestion. Taking into account that we did not specifically include this information in the statistical analysis, we have deleted the data as we have considered irrelevant information.

We have deleted the following sentence:

Of those of working age, 87% of the patients (n=710) had a job considered of risk for developing CTS

Lines 196-222: The information presented in this section appears to represent results rather than discussions. To enhance clarity, it is suggested to refine the results paragraph for improved coherence and subsequently discuss the findings.

Thank you for this comment. We followed your advice and simplified by deleting some of the repeated data already shown in the results section. We only left references to the results needed for discussion comments and comparisons to literature.

Line 266 and Algorithm: As previously emphasized, an algorithm is a consequential outcome of the study and, therefore, should be initially presented as a result. Subsequently, the algorithm can be thoroughly discussed in relation to existing literature. This adjustment will adhere to standard reporting conventions and facilitate a more comprehensive understanding of the research outcomes.

We have included a short methodology for the algorithm reasoning.

We consider that a simple-to-follow algorithm can help general practitioners select patients that would be suitable for ENG, and would help enhance conservative treatment that was poorly used in our series. A recent Cochrane review about splinting considers its use reasonable even with a low degree of certainty. Individualized treatment decisions out of this protocol can be made. American guidelines are very accurate in many procedures but do not establish a clear cut-off for the severity of symptoms or severity of nerve involvement. They do not help with the selection of patients for surgical approach. If we consider clinical suspicion the only requirement for a surgical approach ENG would not be useful in any patient.

As far as we lack information to give a clear surgical cut of, we have decided to delete the treatment part of our algorithm, which is not a main part of our work.

We have clarified in the text as follows:

This algorithm would allow us to guide diagnosis following clinical criteria, one of the strongest predictors for pathological neurography according to our results. It establishes a cut-off regarding symptom severity based on previous definitions (34) and it helps enhance symptomatic treatment since clinical suspicion. According to a recent Cochrane review, even small benefits in patients with splitting seems to justify its use. The benefits would manifest in the long-term use (39). The algorithm could also reduce current ENG demand in patients with a low risk of sensory-motor neuropathy, and this could impact by shortening the waiting time until ENG in those patients at high risk of sensory-motor neuropathy. We also provide a specific and clear cut-off for ENG findings that makes it reproducible for any clinical neurophysiology department, and finally, it could help in surgical decisions. The global clinical relevance should be assessed in future studies.

In summary, the paper demonstrates considerable significance, yet it necessitates enhanced organizational structure. Moreover, I encourage the authors to incorporate references to the American Academy of Orthopaedic Surgeons (AAOS) guidelines on carpal tunnel syndrome (https://www.aaos.org/quality/quality-programs/upper-extremity-programs/carpal-tunnel-syndrome/), a notable omission in the current reference list. An insightful discussion elucidating the authors' findings in light of the AAOS guidelines would contribute significantly to the overall scholarly discourse.

We have included these guidelines in the introduction as well as in the discussion. The guidelines are very complete but lack of assessment of indication for conservative vs surgical treatment.

  1. Reviewer 2 proposed a change in the tittle of the paper, suggesting being more descriptive. We used one of your first sentences, as we understand it summarizes well the study.

Reviewer 2 Report

Comments and Suggestions for Authors

Title:

I suggest that the authors consider incorporating their main finding into the article's title, rather than using a generic title. Doing so would enhance the article's appeal to a broader readership, increasing the likelihood that the average reader will be drawn to read it.

Introduction:

Line 47: “The classic Phalen's definition required sensory alterations restricted to the median nerve area and the positivity of Tinel and Phalen signs (8).” This sentence is unclear, please revise.

Line 54-57: This paragraph contains a repetition of the preceding sentence. Please edit the paragraph to enhance clarity and establish a more logical order.

The introduction would benefit from incorporating references to previous studies. It is important to outline what is currently known about the topic. Additionally, it is crucial to address the significance of the research topic. Why is this study important? The introduction should emphasize the relevance and potential implications of the research findings.

Methods:

 -        Why was the Durkan test not included in the physical examination?

-        Line 97: “Patients were classified according to two subgroups “. Please clarify the methodology employed for classifying patients into the cCTS and aCTS groups.

Results

I suggest revising the results section to improve readability. Consider presenting the results in a table or graph format, while highlighting the most significant findings in the text. This approach will enhance clarity and facilitate a more accessible understanding of the results for the average reader. 

Discussion

 Start the discussion with the main findings of the research.

Avoid repetitions from the introduction and the results.

I find the methodology for developing the algorithm, particularly the treatment decision boxes, unclear. Additionally, given the controversy surrounding the referral for EMG/CT, I am uncertain about how this algorithm would practically impact or alter current medical practices. It would be helpful for the authors to provide clarification on these points. Furthermore, paradoxically, patients with a severe and distinctly evident clinical presentation of carpal tunnel syndrome (CTS) may not necessarily require EMG confirmation. In this subset of patients, the diagnosis is primarily clinical, and surgical intervention may be recommended even in the absence of positive EMG findings After reviewing the study and its conclusions, I am unsure about the extent to which it contributes to existing knowledge and, more importantly, whether it holds practical clinical relevance. Please address these concerns and provide further clarification on the practical implications of the study's findings.

Author Response

Thank you very much for your letter and for the reviewers' comments concerning our manuscript titled: A proposal for neurography referral in patients with carpal tunnel syndrome based on a retrospective study of 797 patients.

We proceed as follows to reply to the reviewers:

Title:

I suggest that the authors consider incorporating their main finding into the article's title, rather than using a generic title. Doing so would enhance the article's appeal to a broader readership, increasing the likelihood that the average reader will be drawn to read it.

Thank you for your suggestion. We have modified the title by borrowing a sentence from reviewer 1 that perfectly resumed the content of our paper.

Introduction:

Line 47: “The classic Phalen's definition required sensory alterations restricted to the median nerve area and the positivity of Tinel and Phalen signs (8).” This sentence is unclear, please revise.

Thank you for your suggestion. We have modified the sentence as follows:

The classical Phalen's definition of CTS requires sensory manifestations in the median nerve sensory territory and the positivity of Tinel’s sign and Phalen’s maneuver (8).

Line 54-57: This paragraph contains a repetition of the preceding sentence. Please edit the paragraph to enhance clarity and establish a more logical order.

We have edited the paragraph to improve comprehension.

The classical Phalen's definition of CTS requires sensory manifestations in the median nerve sensory territory and the positivity of Tinel’s sign and Phalen’s maneuver (8). Other signs and maneuvers have been described for the physical examination of CTS such as inverted Phalen's, or Durkan maneuvers (8,16,17). They all are limited by interobserver variability

The introduction would benefit from incorporating references to previous studies. It is important to outline what is currently known about the topic. Additionally, it is crucial to address the significance of the research topic. Why is this study important? The introduction should emphasize the relevance and potential implications of the research findings.

The most recent study following this purpose was included. We have added another reference. There are not many studies correlating clinical characteristics of patients and neurography results, this is why we consider our study might be useful.

Methods:

 Why was the Durkan test not included in the physical examination?

We thank the reviewer for raising this important issue. Durkan test was not systematically performed in many of our patients, so, as far as it is a retrospective study, we lack this information in our sample.

Line 97: “Patients were classified according to two subgroups “. Please clarify the methodology employed for classifying patients into the cCTS and aCTS groups.

Thank you for you advise, we have clarified the information as follows:

Patients were classified according to two subgroups: "symptomatology compatible with CTS" (cCTS) and "atypical symptomatology for CTS" (aCTS). cCTS were considered patients fulfilling any of the following criteria: 1) Sensory symptoms in median nerve territory, 2) Nocturnal hand symptoms with stiffness or pain that awaken the patient, 3) Sensory symptoms that worsen with manual activity, and improve by flicking, 4) Data from the physical examination; Tinel's sign or Phalen's maneuver (15,17). aCTS were considered patients with a clinical suspicion of CTS referred to ENG evaluation not fulfilling any of the previous criteria, and showing other symptoms, such as isolated pain in the wrist, pain in the first metacarpal joint…

Results

I suggest revising the results section to improve readability. Consider presenting the results in a table or graph format, while highlighting the most significant findings in the text. This approach will enhance clarity and facilitate a more accessible understanding of the results for the average reader. 

Thank you for this suggestion. We have followed your indications and simplify data by adding a table with some information about patients. This allowed us to delete several sentences of the text:

with 70.6% females (n=576) and 29.4% males (n=240).

Ninety-six percent of the patients were right-handed, and only 4% were left-handed.

Of those of working age, 87% of the patients (n=710) had a job considered of risk for developing CTS. We considered an occupational risk  those with repetitive movements of the hands and professional use of computer  55% of the patients presents a clinical symptomatology lasting more than one year. Symptomatology was bilateral in 61% (n=498) of patients and unilateral in 39% (n=318). 

Eight percent of the patients had been treated with a carpal splint before referral. Of those classified as aCTS, less than 4.1% had been treated by splints.

After examination by the neurologist, 57% (n=465) of the patients resulted in Tinel's sign in the right hand and 58% (n=473) in the left hand.  Regarding Phalen's maneuver, 39% of the patients (n=319) had a positive maneuver in the right hand and 34.1% (n=278) in the left hand.

Discussion

Start the discussion with the main findings of the research.

Thank you very much for this suggestion. Following you comment we have modified the text as follows.

We studied demographic and clinical data of our cohort, and analyzed their relationship with the ENG results, focusing on the detection of neuropathy with both sensory and motor involvement (grade 3). The variables with statistical significance for this grade of neuropathy were: typical symptomatology (cCTS), male gender, age, Tinel sign, the positivity of the Phalen maneuver, and the presence of symptoms in the dominant hand and bilateral (table 5).  

Avoid repetitions from the introduction and the results.

We have eliminated repetitions, e.g. “The high prevalence of CTS makes it mandatory a correct management to avoid unnecessary spending of resources.  Neurography has been considered the gold standard for its evaluation.”

We have deleted some of the repeated data showed in the results section and we only left references to the results needed for discussion comments and comparations to literature.

I find the methodology for developing the algorithm, particularly the treatment decision boxes, unclear. Additionally, given the controversy surrounding the referral for EMG/CT, I am uncertain about how this algorithm would practically impact or alter current medical practices. It would be helpful for the authors to provide clarification on these points.

As far as we lack information and our study was not designed for treatment purposes, we have deleted treatment decision boxes in the algorithm.

Furthermore, paradoxically, patients with a severe and distinctly evident clinical presentation of carpal tunnel syndrome (CTS) may not necessarily require EMG confirmation. In this subset of patients, the diagnosis is primarily clinical, and surgical intervention may be recommended even in the absence of positive EMG findings.

In our environment, no patient is operated without EMG performed as it can be helpful in the case of a neurophysiological follow up under several conditions.

After reviewing the study and its conclusions, I am unsure about the extent to which it contributes to existing knowledge and, more importantly, whether it holds practical clinical relevance. Please address these concerns and provide further clarification on the practical implications of the study's findings.

We have included a short methodology for the algorithm reasoning.

We consider that a simple-to-follow algorithm can help general practitioners select patients that would be suitable for ENG, and would help enhance conservative treatment that was poorly used in our series. A recent Cochrane review about splinting considers its use reasonable even with a low degree of certainty. Individualized treatment decisions out of this protocol can be made. American guidelines are very accurate in many procedures but do not establish a clear cut-off for the severity of symptoms or severity of nerve involvement. They do not help with the selection of patients for surgical approach. If we consider clinical suspicion the only requirement for a surgical approach ENG would not be useful in any patient.

As far as we lack information to give a clear surgical cut of, we have decided to delete the treatment part of our algorithm, which is not a main part of our work.

We have clarified in the text as follows:

This algorithm would allow us to guide diagnosis following clinical criteria, one of the strongest predictors for pathological neurography according to our results. It establishes a cut-off regarding symptom severity based on previous definitions (34) and it helps enhance symptomatic treatment since clinical suspicion. According to a recent Cochrane review, even small benefits in patients with splitting seems to justify its use. The benefits would manifest in the long-term use (39). The algorithm could also reduce current ENG demand in patients with a low risk of sensory-motor neuropathy, and this could impact by shortening the waiting time until ENG in those patients at high risk of sensory-motor neuropathy. We also provide a specific and clear cut-off for ENG findings that makes it reproducible for any clinical neurophysiology department, and finally, it could help in surgical decisions. The global clinical relevance should be assessed in future studies.

Round 2

Reviewer 2 Report

Comments and Suggestions for Authors

.